# Cancer-Associated Fibroblasts in Breast Cancer Treatment Response and Metastasis

**DOI:** 10.3390/cancers13133146

**Published:** 2021-06-23

**Authors:** Patricia Fernández-Nogueira, Gemma Fuster, Álvaro Gutierrez-Uzquiza, Pere Gascón, Neus Carbó, Paloma Bragado

**Affiliations:** 1Department of Biochemistry and Molecular Biomedicine, Institute of Biomedicine, University of Barcelona (IBUB), 08028 Barcelona, Spain; gemmafuster@ub.edu (G.F.); peregascon@ub.edu (P.G.); ncarbo@ub.edu (N.C.); 2Department of Biomedicine, School of Medicine, University of Barcelona, 08028 Barcelona, Spain; 3Department of Biochemistry & Physiology, School of Pharmacy and Food Sciences, University of Barcelona, 08028 Barcelona, Spain; 4Department of Biosciences, Faculty of Sciences and Technology, University of Vic, 08500 Vic, Spain; 5Department of Biochemistry and Molecular Biology, Faculty of Pharmacy, Complutense University of Madrid, 28040 Madrid, Spain; alguuz@ucm.es; 6Health Research Institute of the Hospital Clínico San Carlos, 28040 Madrid, Spain

**Keywords:** breast cancer, cancer-associated fibroblasts, therapy resistance, metastasis

## Abstract

**Simple Summary:**

Breast cancer is a major public health problem with a large impact on the life of patients and their families. It is a highly curable disease when detected early, and an inevitably mortal disease when discovered too late. Therapy resistance and metastases are the most critical clinical issues faced by breast cancer oncologists nowadays. It has become evident that interactions between carcinoma cells and tumor microenvironment are an essential part of tumor growth and progression. Cells that support the function of epithelial cells, like cancer-associated fibroblasts (CAFs), contribute to therapy resistance and metastasis via the production of several secreted factors and direct interaction with cancer cells. Here we review the role of CAFs in radiotherapy, chemotherapy, endocrine and targeted therapies resistance. We also highlight the role of CAFs and fibroblasts from metastatic sites in metastasis progression. Finally, we discuss advances and potential therapeutic strategies to target CAFs for overcoming resistance and preventing metastases.

**Abstract:**

Breast cancer (BrCa) is the leading cause of death among women worldwide, with about one million new cases diagnosed each year. In spite of the improvements in diagnosis, early detection and treatment, there is still a high incidence of mortality and failure to respond to current therapies. With the use of several well-established biomarkers, such as hormone receptors and human epidermal growth factor receptor-2 (HER2), as well as genetic analysis, BrCa patients can be categorized into multiple subgroups: Luminal A, Luminal B, HER2-enriched, and Basal-like, with specific treatment strategies. Although chemotherapy and targeted therapies have greatly improved the survival of patients with BrCa, there is still a large number of patients who relapse or who fail to respond. The role of the tumor microenvironment in BrCa progression is becoming increasingly understood. Cancer-associated fibroblasts (CAFs) are the principal population of stromal cells in breast tumors. In this review, we discuss the current understanding of CAFs’ role in altering the tumor response to therapeutic agents as well as in fostering metastasis in BrCa. In addition, we also review the available CAFs-directed molecular therapies and their potential implications for BrCa management.

## 1. Introduction

Breast cancer (BrCa) is the most commonly diagnosed cancer among females and the first leading cause of cancer death in women [1]. Despite the fact that the advancements in therapies and early detection have reduced the mortality rate for BrCa, advanced metastatic BrCa is still considered incurable. BrCa is a heterogeneous and complex disease. In 2000, Perou and Sorlie reported the molecular intrinsic classification of BrCa that distinguished four BrCa subtypes: Luminal A and Luminal B (both expressing the estrogen receptor (ER), Basal-like and HER2-enriched [2,3]. This classification completely altered BrCa clinical management. The main therapeutic approaches to manage BrCa are surgery, chemotherapy, radiotherapy, targeted therapy, and immunotherapy; however, all of them eventually fail, particularly in patients with advanced metastatic disease. Luminal BrCas are treated with endocrine therapy while HER2-enriched BrCas are treated with HER2-targeted therapies such as trastuzumab or pertuzumab. While chemotherapy is still the main treatment for TNBrCa patients, recently, two targeted therapies: olaparib and talazoparib, both Poly(ADP-Ribose) polymerase 1 (PARP) inhibitors, have been approved by the FDA for the management of TNBrCa with germline BRCA mutations [4,5].

Over the past few decades, it has become evident that the complexity of cancer is not only dependent on the intrinsic characteristics of tumor cells but also mainly determined by the crosstalk between tumor cells and various components of the tumor microenvironment (TME). The TME is an ecosystem composed of many cell types (e.g., cancer cells, fibroblasts, immune cells, endothelial cells) located in a complex physicochemical environment. Understanding how this complex ecosystem is organized is critical to further comprehend specific cell population roles in cancer progression from carcinoma in situ to invasive therapy responses and metastasis [6].

Among all components of the TME, fibroblasts are the most abundant cell type, playing an active role in the tumor mass. Fibroblasts are in constant communication with cancer cells, either via direct cell–cell contact or through the secretion of soluble factors able to activate multiple signaling pathways in tumor cells. Unlike normal fibroblasts, which dynamically remodel the ECM, control normal tissue homeostasis and participate in wound healing and senescence, cancer-associated fibroblasts (CAFs) can facilitate tumorigenesis. In particular, BrCa CAFs can represent up to 80% of the tumor mass and are active players in breast tumor initiation and progression [7,8,9,10]. The absence of specific CAFs’ molecular markers has complicated their identification and data comparison between studies. Recently, the analysis of six fibroblast markers: fibroblast activated protein (FAP), integrin β1/CD29, αsmooth muscle actin (αSMA), fibroblast surface protein (FSP1), platelet derived growth factor receptor β (PDGFRβ), and caveolin 1 (CAV1) has allowed the identification of four BrCa CAFs’ subsets that accumulate differentially in normal tissue and in BrCa subtypes, exerting different roles in tumor immunobiology and metastasis [11].

CAFs play multiple versatile functions in cancer, including extracellular matrix remodeling, maintenance of stemness, blood vessel formation and promotion of cancer cell proliferation, migration and invasion, all of which leads to therapy resistance and metastasis formation [12,13]. In recent years, several reports have shown that CAFs play also a critical role in reprogramming the metabolic landscape of tumors [14]. Furthermore, CAFs can also regulate the neighboring immune cells contributing to immune escape of tumors via multiple mechanisms, including secretion of multiple cytokines and chemokines as well as recruitment and modulation of tumor-infiltrating immune cells [15]. The crosstalk between CAFs and cancer cells is mediated by various intracellular and extracellular factors which could potentially be targeted for anticancer therapy.

Improving patient treatment requires a better understanding of the mechanisms leading to resistance to current therapies and metastasis, as well as a deeper comprehension of the role played by the stroma-tumor cells’ crosstalk in these processes. In this review, we focus specifically on the role of CAFs on BrCa progression from ductal carcinoma in situ (DCIS) to invasive carcinoma, therapy resistance and metastasis as well as the strategies and challenges associated with therapeutic regimens targeting the stroma.

## 2. Role of Fibroblasts on Ductal Carcinoma In Situ Progression

DCIS is the most common non-invasive, pre-malignant BrCa lesion and represents up to 25% of all newly diagnosed BrCa patients [16]. DCIS lesions are highly heterogeneous, and each DCIS has its own probability of progressing to invasive ductal carcinoma (IDC), and eventually undergo metastasis [17]. Deciphering the molecular and cellular processes that are occurring in DCIS transition to IDC will shed light on DCIS biology and help identify tumor and stromal biomarkers, able to predict the IDC risk.

The DCIS TME shows a high variability in composition and collagen amounts and in the topological distribution of stromal cells, including fibroblasts [18,19,20,21]. In fact, high collagen deposition and high mammographic density (HMD) are closely related to increased risk of BrCa [22]. The transcriptional profile of fibroblasts present in HMD samples indicated that they are prone to over-activate stress response, inflammation, stemness and C-Jun N-terminal kinase 1(JNK1) pathway and expression patterns similar to those of CAFs in BrCa [23]. 3D co-cultures of fibroblasts and MCF10DCIS.com, a mild aggressive BrCa cell line with low invasive characteristics, have revealed that fibroblasts promote DCIS invasiveness by increasing matrix metalloprotease 14 (MMP14) and MMP9 levels, changing collagen organization and rising BrCa cell proliferation rate [24,25,26]. Moreover, co-injection of MCF10DCIS.com with either normal fibroblasts, CAFs or rheumatoid arthritis fibroblasts resulted in an increased ability to invade the adjacent stroma, suggesting that fibroblasts can facilitate tumor cell invasion, even in the presence of the tumor-suppressing myoepithelial cells [24].

In Basal-like DCIS, the invasive potential seems to rely on the ability of fibroblasts to modify the orientation of collagen fibers [27], with an increased and perpendicular alignment of collagen near DCIS invasive lesions [28,29] compared to the normal collagen organization, parallel to the duct perimeter. HER2-enriched DCIS has also been reported to display perpendicular collagen orientation [28]. However, recurrence risk in DCIS was not associated to the collagen distribution itself, but to the fiber width and density [30]. Interestingly, collagen radial distribution, typical of IDC, is also present in the points of invasiveness in the DCIS, along with collagen perpendicular disposition to the tumor. Therefore, it can be suggested that collagen organization can be informative of the prognosis of DCIS lesions.

Regarding soluble factors, interleukin 6 (IL6) production by CAFs promotes DCIS invasion through the induction of cathepsin B expression in tumor cells [26]. Moreover, it has been demonstrated that primary tumor CAFs enhance invasion of tumor cells, via CAF chemokine (C-X-C motif) ligand 1 (CXCL1) secretion and interaction with C-X-C motif chemokine receptor 2 (CXCR2) in the tumor cells [31]. In DCIS human samples with invasive component, the presence of myofibroblasts positive for urokinase/plaminogen activator (uPA), plasminogen Activator/Urokinase Receptor (uPAR), MMP13, and/or procoagulant factors, such as tissue factor (TF), thrombin and proteinase-activated receptor 2 (PAR2), is related to progression to IDC [32,33,34]. Furthermore, the combination of high expression of Fibroblast activated protein alpha (FAP-α) in fibroblasts and Golgi phosphoprotein 3 (GOLPH3) in BrCa cells can be considered as predictive for DCIS recurrence and progression to IDC [35]. Additionally, CAV1 loss of expression in DCIS fibroblasts has been suggested to predict risk progression to IDC [36], although more studies are needed to validate these results. Therefore, CAFs can facilitate DCIS progression to IDC (Figure 1 and Appendix A).

## 3. Role of CAFs in BrCa Therapy Resistance

### 3.1. Radiotherapy Resistance and CAFs

Radiation therapy is recommended for most people who have lumpectomy to remove BrCa. The goal is to destroy any residual cancer cells that may have remained in the breast after tumor removal and reduce the risk of recurrence [37]. However, radiotherapy (RT) usually results in fibrosis, which may lead to survival and RT resistance of cancer cells [38,39] (Figure 1 and Appendix A).

In the BrCa context, Steer et al. have demonstrated, in 3D co-cultures and in vivo, the protective role of fibroblasts on irradiated cancer cells and tumors [40]. Transforming growth factor β (TGFβ) signaling was found upregulated in the mammary gland after RT, being the main inductor of the stromal compartment activation in in vivo models [41]. Bouquet et al. found that TGFβ inhibition sensitizes BrCa cells to RT [42]. Moreover, different studies have shown that the combination of TGFβ inhibition and RT blocks tumor growth, preventing the increase in circulating TGFβ levels induced by RT, and consequently metastasis, in preclinical studies [43].

Other growth factors have been involved in RT resistance. Hepatic growth factor (HGF) secreted by CAFs activates MET proto-oncogene, receptor tyrosine kinase (MET) receptor in BrCa cells. De Bacco et al. also found that MET inhibition promotes RT response, reducing BrCa cell invasion prompted by CAFs [44]. Furthermore, after ionizing, radiation fibroblasts upregulated β1 integrin expression, together with AKT serine/threonine kinase (AKT) pathway activation, leading to RT protection in vivo [45]. Boelens et al. further showed that the crosstalk between CAFs and BrCa cells drives chemo- and radiotherapy resistance through an exosome transfer from the stromal population. As a result, Notch receptor 3 (NOTCH3) and Signal transducer and activator of transcription-1 (STAT1)-antiviral pathways were activated, inducing a RT-resistant stem cell-like phenotype [46]. Besides, radiation induces senescence on irradiated fibroblasts, which produce IL6, CXCL8/IL8 and osteopontin (OPN), all pro-tumorigenic factors related with stromal-mediated RT resistance [47]. Senescent CAFs also improved BrCa cells’ proliferation in an AKT pathway-dependent manner, which also conferred RT resistance [48]. On the basis of these observations, it could be argued that targeting CAFs during RT could improve BrCa RT response (Figure 1 and Appendix A).

### 3.2. CAFs and Chemotherapy Resistance in Triple Negative BrCa

TNBrCa or basal subtype represents around 15% of BrCa lesions diagnosed and is characterized by the expression of basal-related genes and the absence or low levels of ER, PR and, HER2 [2]. This BrCa subtype encompasses a wide and heterogeneous group of lesions and is generally associated with a poor prognosis due to the lack of appropriate targeted therapies [49]. Therefore, the main clinical challenge in TNBrCa is to design new therapies to reduce metastasis and improve patient’s quality of life and survival.

Concerning CAFs contribution to therapy-resistance, a 6-CAFs’ genes signature (CXCL2/MIP2 and its human homologous CXCL8/IL8, MMP1, Retinoic Acid Receptor Responder 1 (RARRES1), fibroblast growth factor 1 (FGF1), and CXCR7) has been related to taxotere treatment resistance in an in vitro model using TNBrCa cells [50]. In addition, EDALINE phase I clinical trial results suggest that blocking hedgehog (Hh) stimuli on CAFs, by smoothened inhibitors (SMOi) treatment, sensitizes TNBrCa to docetaxel and leads to survival amelioration [51]. Moreover, inhibition of TGFβ production by CAFs with the anti-fibrotic drug pirfenidone reduced BrCa tumor growth and also lung metastasis when combined with doxorubicin, acting in a synergistic way [52]. Combination of a hydrogel assembled to losartan, an inhibitor of angiotensin-II that blocks the TGFβ pathway, sensitizes TNBrCa cell lines to doxorubicin-loaded liposomes facilitating the reduction of chemotherapy resistance [53].

On the other hand, under chemotherapy pressure, interferon β1 (IFNβ1) secretion by cancer cells stimulates the transcription of pro-inflammatory cytokine genes in stromal fibroblasts, by which CAFs acquire an anti-viral state, essential for the recovery and resistance acquisition of BrCa cells after chemotherapy. Indeed, high expression of tumor IFNβ1 in TNBrCa patients correlated with an aggressiveness signature [54]. Another interesting paracrine relationship between cancer cells expressing platelet derived growth factor (PDGF) and CAFs positive for PDGF-receptors was described in basal-like BrCa patients. PDGF abrogation in the tumor produces a phenotype shift into a positive hormone-receptor subtype, sensitive to endocrine therapy, leading to a more therapeutically affordable BrCa subtype. These results also suggest that BrCa subtype control and plasticity can be exerted by fibroblasts and cancer cells interactions [55].

Furthermore, the interaction between cancer cells, immune cells, mesenchymal stem cells (MSC), and CAFs led to the release of CXCL1, CXCL2/MIP (homologous to CXCL8/IL8 in humans) (from TNBrCa cells), CXCL16 (from CAFs), and C-C motif chemokine ligand 5 (CCL5) (from CAFs and MSCs), resulting in a more reactive stroma, inducing pro-tumoral actions, including chemotherapy resistance [56,57,58,59,60,61]. This pro-resistance context attracts tumor-associated macrophages and stimulates the endothelial cell population through the release of CXCL8/IL8 and CCL5 by TNBrCa cells. All these results suggested that therapeutically targeting CXCL8/IL8 and CCL5 could be useful in TNBrCa patient’s treatment [62,63]. Expression of CXCL14 or Programmed death-ligand 1 (PD-L1) by stromal CAFs has also been associated with shorter or better survival, respectively, in TNBrCa patients [64,65], and could be considered as independent prognosis markers in TNBrCa patients. Therefore, these data emphasize that there is a strong chemotherapy-resistance dependence of tumor-stromal interactions in TNBrCa, particularly evident between fibroblasts and cancer cells (Figure 1 and Appendix A), highlighting the importance of targeting stroma and tumor cell interactions as a promising clinical therapeutic strategy in preventing drug resistance in TNBrCa patients [66].

### 3.3. CAFs and Hormonal Therapy Resistance in Luminal BrCa

Luminal BrCa represents 70% of all BrCas and expresses hormonal receptors, hence it is considered sensitive to endocrine therapy, especially ER-targeted [67]. The prognosis for early luminal BrCa patients is particularly good, however, up to 50% of patients relapse and die from metastatic disease [67]. The treatment of patients with ER-positive BrCa is based on ER modulators such as tamoxifen and/or aromatase inhibitors. Tamoxifen has become the drug of first choice in patients with luminal BrCa. Nevertheless, it is estimated that approximately 45% of women do not respond to tamoxifen (de novo resistance) [68], whereas acquired resistance to the drug develops ultimately in all tamoxifen-receiving patients. Therefore, although tamoxifen was hailed as a major breakthrough in the management of patients with hormone-dependent BrCa, development of resistance to the treatment poses a serious clinical problem [69].

Endocrine-therapy resistance can likewise be linked to the biological action of luminal BrCa CAFs. They express a G protein-coupled estrogen receptor 1 (GPER), which suggest that they are also sensitive to estrogens. CAFs expressing GPER can activate the epidermal growth factor receptor (EGFR) and extracellular signal-regulated kinase 2 (ERK) signaling which leads to (i) estradiol production by CAFs and endocrine therapy resistance [70] and (ii) cancer cell integrin β1 activation by increased CAF-secreted fibronectin [71], which drives EMT and the tamoxifen-resistant phenotype. In turn, CAFs can also become resistant to hormonal therapy. BrCa-resistant cells secrete TGFβ1 that induces ERK activation on CAFs and promotes their resistance [72]. In addition, activation of phosphatidylinositol-4,5-bisphosphate 3-kinase (PI3K)/AKT in BrCa cells increases CAFs’ GPER translocation to the membrane and induces the activation of PKA/cAMP responsive element-binding protein (CREB) signaling, which activates the Warburg effect in CAFs. This increases pyruvate and lactate levels, metabolites used by cancer cells to fuel their mitochondrial metabolism and confer drug resistance to hormonal- and HER2-targeted therapies, as well as chemotherapy [73]. CAFs-mediated metabolic reprograming of BrCa cells to decrease sensitivity to endocrine therapy has been proposed by others. For instance, CAFs’ secretion of lactate and ketone bodies increases mitochondrial activity in BrCa cells and leads to tamoxifen resistance [74]. A recent report has shown that BrCa cells promote neighboring CAFs’ hypoxia, which leads to autophagic degradation of CAFs CAV1 and activation of the Warburg effect in them. These changes contribute to protect BrCa cells from endocrine therapy-induced apoptosis and autophagy, by providing nutrients to fuel BrCa cell metabolism [75]. In fact, BrCa patients whose stroma is negative for CAV1 expression develop resistance to tamoxifen and have worse prognosis [76].

Endocrine therapy resistance is also driven by CAFs’ secretion of growth factors. For instance, HGF secretion by CAFs leads to MET upregulation in ER-positive BrCa cell lines, which leads to increase migration, invasion, and resistance to fulvestrant through SRC proto-oncogene, non-receptor tyrosine kinase (SRC), AKT, and ERK1/2 activation [77]. FGF1 has also been shown to induce hormonal therapy resistance through activation of fibroblast growth factor receptor 3 (FGFR3) that causes PI3K/AKT and ERK1/2 induction by phospholipase C γ (PLCγ) [78]. Additionally, FGF7 can block tamoxifen effect through activation of FGFR2, which enhances ER degradation by the proteasome. FGF7/FGFR2 effect was mediated through activation of PI3K/AKT signaling and the upregulation of B-cell lymphoma 2 (Bcl-2) expression in BrCa cells [79].

Several other reports have linked CAFs-elicited resistance to tamoxifen with activation of PI3K/AKT and/or Ras/Raf/mitogen activated protein kinase kinase 1 (MEK1)/ERK1/2 [80,81,82]. Fibronectin, through interaction with Integrin β1, and soluble factors secreted by CAFs, activate EGFR and MMPs which leads to induction of PI3K/AKT and ERK1/2 in BrCa cells, protecting them from tamoxifen-induced cell death [83]. AKT and ERK1/2 hyperactivation stimulates ER phosphorylation which promotes tamoxifen resistance [80]. Moreover, fibroblasts expressing amyloid beta precursor protein binding family A, member 3 (MINT3) upregulate L1 cell adhesion molecule (L1CAM) which leads to ERK1/2 activation in BrCa cells through integrin α5β1, promoting tumor growth and resistance [81]. Stromal cells have also been shown to elicit hormonal therapy resistance through inhibition of insulin-like growth factor binding protein 5 (IGFBP5) which induces Bcl-3 upregulation and activation of NF-κB, driving resistance to fulvestrant [84].

CD146 is a cell surface marker that characterizes two different subtypes of fibroblasts in BrCa tumors. CAFs negative for CD146 foster the upregulation and activation of several tyrosine kinase receptors (TKRs), such as EGFR, HER2 and insulin-like growth factor receptor (IGF1R), and the inhibition of ER in BrCa cells which leads to tamoxifen resistance [85]. Other cell surface markers have also been linked to CAFs-mediated tamoxifen resistance. For example, a subpopulation of CAFs positive for CD63 has been shown to elicit tamoxifen resistance through secretion of exosomes enriched in miR-22, that activates ERα and phosphatase and tensin homolog (PTEN) promoting resistance. Furthermore, CD63 induces STAT3 activation to maintain the phenotype and function of CD63+ CAFs [86]. Other microRNAs have recently been associated with CAFs-mediated hormonal therapy resistance. It has recently been described that CAFs exosomes or microvesicles containing miRNA-221 can facilitate hormonal therapy resistance by promoting cancer stem cell (CSC) formation and ER downregulation, which leads to tumor growth and resistance to fulvestrant-induced apoptosis [87]. Altogether, these findings indicate that BrCa CAFs mediate resistance to endocrine therapy through several molecular mechanisms and mediators (Figure 1 and Appendix A), each of them representing an intriguing target to be explored to re-sensitize BrCa cells to hormonal therapy.

### 3.4. CAFs- and HER2-Targeted Therapy Resistance in HER2-Enriched BrCa

Around 15% to 20% of all breast tumors belong to the HER2 positive (+) subtype, with an aggressive phenotype and worse clinical course [2,88]. HER2+ subtype is characterized by *ERBB2* amplification, causing an activation of the HER2 pathway, initiating cell proliferation, promoting cell survival, and driving metastasis through different pathways such as the RAS and the PI3K–PKB(AKT)–MAPK pathways [1]. Four targeted drugs are already approved for treatment of these patients, being trastuzumab, pertuzumab, lapatinib, and ado-trastuzumab emtansine (TDM1), the commonly recommended anti-HER2 target agents with excellent therapeutic effects [89]. However, the development of resistance to these treatments has reduced their expectations in terms of overall clinical benefit in the metastatic setting, representing the principal impediment concerning better long-term patient survival [90,91,92,93].

As mentioned before, BrCa treatment is often unsuccessful due to the protection from the tumor microenvironment. Zengh and colleagues recently designed a nano-theranostic platform to battle this difficulty. HER2-DSG NPs (HER2-Doxorubicin-superparamagnetic iron oxide nanoparticles) were able to actively identify and target HER2-positive tumor cells and doxorubicin release. Moreover, NPs enhanced immunogenic cell death effect in vitro/in vivo, activating immune response and reducing CAFs’ population, increasing NPs access to cancer cells and leading to a great antineoplastic effect [94].

During the last years, different studies have highlighted the vital contribution of breast CAFs in the resistance mechanisms to anti HER2-targeted therapies. An interesting study by Nguyen et al. used recent advances in microtechnology of “organ-on-chip” to reproduce ex vivo the complexity of the tumor microenvironment ecosystems integrating the main populations of BrCa stroma: immune cells, endothelial cells and fibroblasts, together with BT-474 HER2+ BrCa cells, in the absence or presence of trastuzumab. They found a pro-invasive effect of CAFs as well as an antagonic immunomodulation effects of CAFs and trastuzumab in the HER2+ tumor ecosystem [95]. Marusyk and colleagues have defined that close physical contact between CAFs and BrCa cells protects BrCa cells from lapatinib and other chemotherapeutic agents such as taxol and doxorubicin [96]. Interestingly, this protection was not restricted to CAFs; stromal fibroblast from normal breast tissues and from brain metastases displayed similar effects [96].

Furthermore, Lin et al. found that increasing matrix stiffness was related with a reduction in lapatinib response, identifying fibronectin and type IV collagen as main contributors to therapy resistance, which was conditional on nuclear translocation of YAP transcription factor [97]. In vivo, antineoplastic effects of lapatinib were improved when treating with a recombinant human hyaluronidase (PEGPH20), which is currently being evaluated in multiple clinical trials under combined treatment settings [96,98].

On the other hand, paracrine communication between BrCa cells and CAFs has been shown to play a key role in HER2-targeted therapy resistance as well. We have recently described that soluble fibroblast growth factor 5 (FGF5) produced by the neighboring CAFs activates FGFR2 in HER2+ BrCa cells which in turn promotes c-Src-mediated HER2 transactivation. Additionally, we have demonstrated that the combination of anti-HER2 targeted therapies with FGFR2 inhibitors is able to overcome the resistance to trastuzumab and lapatinib in vitro and in vivo [99]. Additionally, in a cohort of trastuzumab-treated patients, high FGF5 stromal expression and high levels of HER2 phosphorylation define a group of patients who will not benefit from trastuzumab neoadjuvant treatment. These data led us to propose FGF5 and phospho-HER2 as potential biomarkers to predict resistance to HER2-targeted therapies [99]. In line with our results, Akhand et al., using HER2+ patient derived xenografts (PDX), obtained and characterized TDM1-resistant cells, also unable of responding to lapatinib and afatinib, but with an acquired sensitivity to a covalent FGFR kinase inhibitor, strongly supporting therapeutic combination of TDM1 with FGFR inhibitors in HER2+ BrCa treatment [100]. The role of FGFs and FGFRs in CAFs’ crosstalk with HER2+ BrCa has been also addressed by Hanker et al. [101]. By generating HER2-amplified BT-474 xenografts resistant to the combination of lapatinib and trastuzumab, they found increased copy number for FGF3, FGF4, and FGF19 genes in resistant tumors [101]. Co-treatment with the FGFR inhibitor lucitanib reversed collagen deposition in tumor stroma, as well as fibronectin, and increased αSMA staining (markers of activated stromal fibroblasts) in resistant tumors, suggesting that changes in the microenvironment are in part dependent on the FGFR pathway [101].

By using microenvironment microarrays (MEMA) [102], Watson et al. found that single growth factors like HGF or neuregulin 1 β (NRG1β), highly expressed by breast fibroblasts, were linked with lapatinib resistance in different HER2+ BrCa cell lines [103]. Independent HER2+ models also showed HGF ability to reestablish PI3K and Ras/Raf/MEK/ERK1/2 pathway activities that were initially lost under treatment with lapatinib, thereby stimulating drug resistance [104].

CAFs-derived cytokines have also been related with trastuzumab resistance. IL6 secreted by CAFs activates the STAT3 pathway and downregulates PTEN, inducing CSC population expansion [105]. Moreover, a latest report by Zervantonakis et al. has described that fibroblast-secreted factors induce survival in response to lapatinib, through mammalian target of rapamycin (mTOR) signaling activation and enhanced antiapoptotic protein levels. In accordance, mTOR, Bcl-XL or myeloid cell leukemia sequence 1 (MCL1) inhibitors restore lapatinib drug sensitivity [106]. Undoubtedly, these studies underline the idea that co-targeting the tumor and the stroma poses a great opportunity to improve current anti-HER2 therapies and to overcome treatment resistances (Figure 1 and Appendix A).

## 4. CAFs and BrCa Metastasis

Metastasis is the principal cause of cancer death. As previously exposed, the crosstalk between stromal and tumor cells sustains cell survival, cell growth and therapeutic resistance. In this section, we will examine evidence involving fibroblast role in BrCa metastasis. We will analyze both the role of primary tumor CAFs in mediating the first steps of the metastatic cascade, such as epithelial to mesenchymal transition (EMT), migration and invasion, and the role of the secondary organs’ fibroblasts in the regulation of disseminated tumor cells’ (DTCs) colonization, dormancy and survival.

### 4.1. Contribution of CAFs from Primary Tumor Stroma to Metastasis Progression

Metastasis is a complex process that needs assistance from neighboring cells to be more efficient [107]. Hasabe et al. have shown that the existence of fibrotic foci correlates with nodal and distant organ metastasis in patients with IDC [108].

CAFs have been described to contribute to BrCa cells invasion by releasing growth factors (epidermal growth factor (EGF), FGF2 [109], PDGF, vascular endothelial growth factors (VEGFs), insulin-like growth factor (IGF) [110], HGF, tumor necrosis factor (TNF), CXCL12/ stromal cell-derived factor 1 (SDF1) [111,112]) as well as cytokines and chemokines CCL8 [113], CXCL16 [61], IL6 [26,114], IL4 [115], CXCL1 [116], CXCL14 [64], CCL5) [117,118], CXCL8/IL8 and CCL2 [62], IL32 [119]) which influence BrCa cell motility [120]. In a recent study Suh et al. evidenced that CAFs secretion of FGF2 was sufficient to enhance MDA-MB-231 cells growth, migration and invasion via FGFR1 signaling [109]. Dvorak et al. showed that CXCL12/SDF1 secretion induces mammalian Diaphanous-related formin-2 (MDia2) deregulation, leading to breakdown of the F-actin cytoskeleton and a subsequent increase of BrCa cell motility [112]. Moreover, αSMA+ fibroblasts produce high amounts of CXCL12/SDF1 and IGF1, which select cancer cells with high Src activity prone to colonize CXCL12/SDF1-rich bone marrow microenvironment, further suggesting a potential role of primary tumor CAFs in educating BrCa cells to metastasize to the bone microenvironment [121]. In addition, fibroblasts at tumor margins secrete higher levels of CCL8, generating a gradient between the epithelium, the stroma and the periphery that attracts BrCa cells to migrate and disseminate [113]. Besides, high stromal expression of CXCL14 is significantly associated with shorter recurrence periods [64]. A tumor necrosis factor alfa (TNFα) and IL1β enriched environment promotes metastasis in TNBrCa cells through the shift to a metastatic phenotype in both stromal and cancer cells, leading to an increased expression of pro-metastatic and immune-evasive chemokines, CXCL8/IL8, CCL2 and CCL5, among others, induction of angiogenesis and enhanced invasive and migration abilities of TNBrCa cells [62,122,123,124,125,126,127].

Several reports have also defined the importance of the release by CAFs of adipokines such as leptin in promoting the malignant aggressiveness of BrCa cells [128]. Giordano et al. described leptin as a regulator of the crosstalk between the stromal and BrCa cells. Leptin production by CAFs induce mammosphere formation, suggesting leptin can promote BrCa cells stemness [129]. Moreover, the inhibition of leptin with a synthetic farnesoid X receptor (FXR) agonist GW4064, reduces the pro-tumorigenic effect of CAFs in BrCa [130]. Other groups have also described an important role of leptin in potentiating BrCa cells migration and angiogenesis, being also involved in EMT [131]

Luga et al. also reported that CAFs produce exosomes that boost the motility and metastatic potential of BrCa cells by activating Wintless-INT family (Wnt) signaling [132]. Otherwise, BrCa exosomes such as miR-146a regulate the activation of Wnt/β-catenin signaling pathway on fibroblasts, which become CAFs contributing to invasive and metastatic abilities of BrCa cells [133]. Conversely, CAFs-derived exosomes induce proliferation and metastasis through the miR-500a-5p transference to luminal and basal-like BrCa cells. miR-500a-5p interacts with and blocks the effects of USP28, considered by the authors as a possible BrCa tumor suppressor [134]. Another work demonstrates that CAF exosomes can release miR-21, -143, and -378 that enhance important aggressive cancer hallmarks such as stemness, EMT and the ability to grow in an anchorage-independent way in TNBrCa cells [135].

One of the key steps in the metastatic process is cancer cell EMT and CAFs have been described to secrete growth factors such as transforming growth factor beta 1 (TGFβ1), EGF, PDGF and HGF that will promote EMT in primary tumor cells [107,120,136,137]. Likewise, the increased OPN secretion by TIAM rac1 associated GEF 1 (TIAM-1)-defective CAFs stimulates BrCa EMT, stemness and invasion [138]. Recently, Wen et al. have described that CAFs secrete IL32 that binds integrin β3 on BrCa cells resulting in p38 MAPK pathway activation, further enhancing the expressions of fibronectin, N-cadherin and vimentin [119]. In addition, it seems that activation of autophagy and expression of FAP-α in CAFs can activate EMT in cancer cells via Wnt pathway, resulting in an increase in cell migration, invasion and proliferation [139,140]. An interesting study describes that CAFs expression of phosphatidylinositol-4,5-bisphosphate 3-kinase catalytic subunit delta (PIK3Cδ) fosters TNBrCa cell metastasis, controlling mainly invasion through the inhibition of tumor suppressor genes such as Nuclear Receptor Subfamily 4 Group A Member 1 (*NR4A1)* [141]. Recently, Matsumura and colleagues have established that CAFs-secreted factors control the epithelial/mesenchymal plasticity of distinct metastatic tumor clusters, which comprise two different cell populations, one with a more epithelial phenotype and another with a mixed epithelial-mesenchymal phenotype. Both populations collectively contribute to enhanced tumor cell cluster formation and metastatic seeding in secondary organs [142]. Choi et al. have demonstrated, using three-dimensional in vitro models, that CAFs can facilitate transmigration of BrCa cells through the blood–brain barrier by increasing their expression of α5β1 and αvβ1 integrin and MET [143].

It is accepted that fibroblasts contribute to the remodeling of the surrounding stroma, which is a prerequisite for cancer cells’ invasion and metastasis. In vitro and in vivo studies have indicated that CAFs promote BrCa cell invasion and metastasis by secreting MMPs, such as MMP1, -2, -3, -7, -9, -13, and -14 [136,144,145]. Moreover, different research groups have observed that CAFs may generate tracks for tumor cells to go forward. For instance, Syndecan-1 expressing fibroblasts modify the collagen fibers in the extracellular matrix to form parallel structures that can be utilized by BrCa cells as a guide [146]. Ahirwar and colleagues showed that CAFs secreted CXCL12/SDF1, generating endothelial instability and hyper-permeable vasculature, facilitating the escape of tumor cells from the primary tumor to distant organs [147]. Other studies have additionally indicated that CAFs adopt a desmoplastic program with altered cell adhesion properties which pushes the progression of metastasis through induction of mechanical pressure on cancer tissue [148]. These studies highlight the important role of CAFs in mediating BrCa cell migration, EMT and invasion (Figure 2 and Appendix A), hence suggesting stroma-targeted therapies might be an interesting approach to address metastatic disease.

### 4.2. Contribution of Fibroblasts from Metastatic Sites to Metastasis Formation

Once disseminated, tumor cells arrive to metastatic target organs where they will start interacting with stromal cells of the target organ. This new dialogue between DTCs and stromal cells will determine DTCs survival, activation of dormancy or the shift to a proliferative phenotype, hence regulating metastasis progression [149]. Gui et al. recently published a comparative study with BrCa fibroblasts from primary site (pCAFs), metastatic organs (mCAFs) and normal fibroblasts. They showed that mCAFs display more marked pro-tumorigenic effects, also conferring greater protection to cytotoxic drugs and improving metastatic capacity. All these differential capabilities of mCAFs were associated in part with their higher secretion of IFNβ and IGF2, which in turn activates IGF1R signaling [150]. BrCa DTCs usually colonize lungs, bone marrow and brain [151]. It has been described that communication between primary tumors and their potential metastatic sites starts even prior to the arrival of DTCs to the target organ. In the next paragraphs, we will focus on the accommodation of the stroma of the target organs and its role on DTCs fate.

During the early dissemination phase to the lungs, different primary tumor-derived mediators, such as growth factors, cytokines, extracellular matrix (ECM)-remodeling enzymes or extracellular vesicles and exosomes, have the ability to modify the pulmonary parenchyma creating a welcoming microenvironment for the tumor cells to seed, proliferate and survive [152]. Secretion of TNFα and TGFβ, along with vascular endothelial growth factor A (VEGFA), induce the expression of S100 calcium binding protein A8 (S100A8) and S100 calcium binding protein A9 (S100A9) in lung fibroblasts to prepare premetastatic niches [153]. S100A8 and S100A9 stimulate the recruitment of Mac-1+ myeloid cells to the lung, resulting in the secretion of factors able to stimulate migration like TNFα, CXCL2/MIP2 and TGFβ, as well as ECM remodeling [154]. A study by Raz et al. further demonstrated that bone marrow-derived stromal cells (BM-MSCs) are the precursors of a significant fraction of CAFs that are present in both primary breast and metastatic lung lesions [155], which are able to trigger a distinctive inflammatory profile, inducing pro-metastatic characteristics like angiogenesis and supporting metastasis to the lungs [155].

Exosomes derived from PT cancer cells play a critical role in transforming the lung microenvironment. A thorough study by Hoshino and colleagues revealed an organ specificity of exosome biodistribution that matched the organotrophic distribution of the cell line of origin [156]. Interestingly, they also found that lung-tropic exosomes, expressing α6β4 and α6β1 integrins, predominantly co-localized with S100 calcium binding protein A4 (S100A4)-positive fibroblasts and surfactant protein C-positive epithelial cells in the lungs, enhancing lung metastases [156]. Accordingly, integrin α6β4 targeting reduced lung metastasis [156]. Interestingly, Fong et al. have shown that exosomal miR-122 is able to reprogram lung niche cells to a lower glucose demand, by down-regulating pyruvate kinase activity. This metabolic reprogramming results in an increased DTCs glucose availability to effectively drive their proliferation [157].

Additionally, the contribution of BrCa-secreted exosomal long non-coding RNAs (LncRNAs) on lung pre-metastatic niche education has been described. These LncRNAs stimulated lung fibroblast proliferation and migration by targeting different genes and signaling pathways, including TGFβ, pentose phosphate, Hh, metabolic, and complement and coagulation cascade pathways [158]. Medeiros et al. found that extracellular vesicles (EVs) from TNBrCa cells were able to stimulate the expression of pre-metastatic ECM markers in lung fibroblasts [159].

During early metastatic colonization, lung DTCs reprogram resident lung fibroblasts to create a more favorable environment for proliferation and survival. For instance, DTCs decrease miR-30 family members’ expression in lung fibroblasts, which inhibit MMP9 expression, increasing vascular permeability and consequently favoring metastasis [160]. BrCa DTCs also induce the expression of αSMA in resident fibroblasts in the lung. Activated fibroblasts direct extracellular matrix remodeling by the secretion of fibronectin and LOXL2, creating a more tolerant microenvironment for DTCs to survive and outgrow [161]. Moreover, cancer cells are able to educate fibroblasts to produce periostin and Tenascin C (TNC), enhancing metastatic colonization of the lung [162]. Periostin expression in lung fibroblasts may be modulated by cancer cells through the secretion of transforming Growth Factor Beta 3 (TGFβ3). Fibroblasts in turn recruit Wnt ligands to present to cancer cells and promote DTCs stemness by inducing Wnt signaling, enhancing lung colonization [152]. Periostin also fosters the lung recruitment of MDSCs at the metastatic site, especially in ER-negative tumors, essential for achieving an immunosuppressive premetastatic niche environment. In addition, periostin can bind to TNC, anchoring it to ECM components such as fibronectin and type I collagen [163]. DTCs-derived TNC supports early metastasis outgrowth, until the lung stroma takes over as the main source of TNC. TNC can also induce the secretion of growth factors, like EGF or FGF, and interact with fibronectin, heparin-sulfate proteoglycans, fibrinogen, integrins, MMPs, and EGFR [164]. Additionally, high thrombospondin 2 (THBS2)-expressing cancer cells are also more efficient on activating lung fibroblasts, which subsequently induce the transition of tumor cells into a more epithelial phenotype [165]. In addition, Shu et al. have described a new role of the complement system C3a-C3aR on metastasis promotion by modulating CAFs activation, enhancing pro-metastatic cytokine secretion via the activation of PI3K-AKT signaling [166].

Bone is the main target organ for HER2+ BrCa metastasis, representing the major cause of pain and severe complications for these patients, including fracture, spinal cord compression or hypercalcemia [167]. Kang et al. have described a 6-gene bone metastasis signature in BrCa cell lines using mouse models. Interestingly, one of the genes was FGF5 that our group has recently described to be implicated in HER2+ targeted therapy response [99,168]. It has also been reported that BrCa cells in the bone marrow display higher expression of nanog homeobox (NANOG) *OCT4* and *SOX2* in the presence of MSC-derived CAFs and OPN [169], enhancing BrCa cells’ stemness. In turn, vimentin-expressing FAP+ fibroblasts produce miR-221 containing exosomes that further contribute to maintain the stem phenotype in favor of metastasis [87]. Furthermore, DTCs in the bone cavity produce abundant C-C motif chemokine ligand 4 (CCL4) as well, to attract type I collagen-expressing α-SMA+ fibroblasts expressing C-C motif chemokine receptor 5 (CCR5). Accumulated fibroblasts support cancer cells’ growth by enhancing connective tissue growth factor signals [170].

Regarding BrCa liver metastasis and the potential role of fibroblasts aiding the process, it has been defined that OPN promotes tumor progression via the transformation of MSC into CAFs in the liver as well. OPN was responsible for MSC CCL5 production. Moreover, the MSCs recovered from liver metastatic sites present CAF markers’ expression (α-SMA, TNC, CXCL12/SDF1, and fibroblast-specific protein-1) as well as MMP-2 and MMP-9 [171].

All these studies underline the essential role of resident fibroblasts in reprogramming the metastatic niche to favor DTCs seeding and survival at secondary organs (Figure 3 and Appendix A). These insights could help design therapies targeting the cancer cell-stromal crosstalk of primary tumors and metastases, likely improving the clinical management and outcome of the advanced BrCa patients.

## 5. Targeting CAFs to Prevent BrCa Progression

As we have reviewed here, CAFs are major players in driving BrCa progression by affecting cancer cell invasion and colonization ability and its therapeutic response. Their diverse tumor-promoting functions, combined with their genetic stability and relative abundance among stromal cells, make them an appealing therapeutic target. Here, we will briefly highlight the major advances and challenges in the development of CAFs-targeted therapies in BrCa. The therapeutic approaches to target stromal cancer cells in BrCa can be categorized into three main lines: direct targeting of the ECM, targeting of CAFs and targeting of profibrotic signaling [172].

Direct targeting of the ECM has focused on the depletion of ECM molecules (collagen and hyaluronic acid (HA) mainly), and it has been successfully pursued using collagenase and/or relaxin enzymes as well as other approaches [172]. For example, PEGPH20 has been successfully used in BrCa cellular models [173], and it is known that it sensitizes HA-rich tumors to anti-PD-L1 immunotherapy in murine syngeneic breast cancer models.

The second antifibrotic strategy focuses on the elimination of CAFs as the main source of the ECM components. This therapeutic approach takes advantage of known CAFs markers such as αSMA, FSP1, and FAP-α to identify and eliminate them [172]. A BrCa phase I clinical trial testing the effects of the construct IL-2v targeting FAP (RO6874281) administered either alone or in combination with trastuzumab or cetuximab (ClinicalTrials.gov; Identifier: NCT02627274) is currently undergoing. However, this strategy still faces numerous obstacles, like unspecific markers and heterogeneity among the CAFs population. Interestingly, a specific population of BMDCs, granulin-expressing BMDCs, are recruited to metastatic niches and induce the tumor-promoting function of mammary fibroblasts. It has been suggested that targeting these granulin-expressing cells could be exploited as anticancer therapy to avoid CAFs support of tumor progression [174]. Moreover, all-trans retinoic acid (ATRA) treatment is able to reprogram activated CAFs, restoring their quiescence in pancreatic cancer [175], and decreasing malignant transformation of ductal carcinoma breast cancer cells [176].

Interfering with the TGFβ or Hh signaling pathways have also been successfully tested in BrCa. TGFβ signaling blocking, by either soluble TGFβ type II receptor or TGFβ neutralizing antibody, significantly decreased tumor growth and metastasis in BrCa [177]. Remarkably, TGFβ signaling can also be blocked through angiotensin receptor inhibitors (ARB), such as losartan, which enhanced delivery of chemotherapy and improved overall survival in BrCa models [178]. Pirfenidone (PFD), an anti-fibrotic agent as well as a TGFβ antagonist, has also been effectively examined in TNBrCa patient-derived xenograft models [66]. Several Hh pathway inhibitors have also been tested, being IPI-926, the one with the best results [179]. Furthermore, focal adhesion kinase (FAK) signaling has also been related with the formation of fibrotic tumor microenvironment [172] and appears as a druggable target, not only in tumor cells but also in the tumor microenvironment [180]. The stromal targets list in BrCa is continually growing, as do newer therapies against stromal components [181,182]. For example, in vitro and in vivo approaches against the phosphodiesterase PDE5 (Sildenafil) or the nuclear receptors FXR (GW4064) have shown promising results due to their relevant role in CAFs biology and pro-tumoral effect [130,181,183,184].

These therapeutic strategies targeting CAF/stroma-specific pathways are already being evaluated in several clinical trials on BrCa, or other solid tumors. Most of them combine classical strategies with CAFs-targeted therapies compounds to successfully improve cancer therapy. The main targets are the following: TGFβ-pathway (fresolimumab (NCT01401062) or galunisertib (NCT02304419)), Hh (vismodegib (NCT02694224) or LDE225 (NCT02027376, NCT02694224)), Notch (MK0752 (NCT00645333)), FGFR-FGFR ligand (Dovitinib (NCT01548924), AZD4547 (NCT01791985), FAP (RO6874813 (NCT02558140)), Hyaluronic acid in pancreatic cancer (PEGPH20 (NCT01453153)) and Multitargets ((Dasatinib (NCT00546104), lucitanib (NCT02053636), Famitinib (NCT04733417, NCT01653574, NCT04129996), Motesanib (NCT01349088).

## 6. Conclusions

In this review, we have summarized the most recent and significant findings on the role of CAFs in BrCa therapy resistance and metastasis. The data revised here emphasizes the role of CAFs as novel, promising a therapeutic target in BrCa suggesting that combination strategies aimed at targeting both BrCa cells and the surrounding microenvironment may enable more efficient therapeutic responses. Furthermore, the identification of novel and specific biomarkers of CAFs will be priceless for accelerating the translation into the clinic of CAFs-targeted therapies.

## Figures and Tables

**Figure 1 cancers-13-03146-f001:**
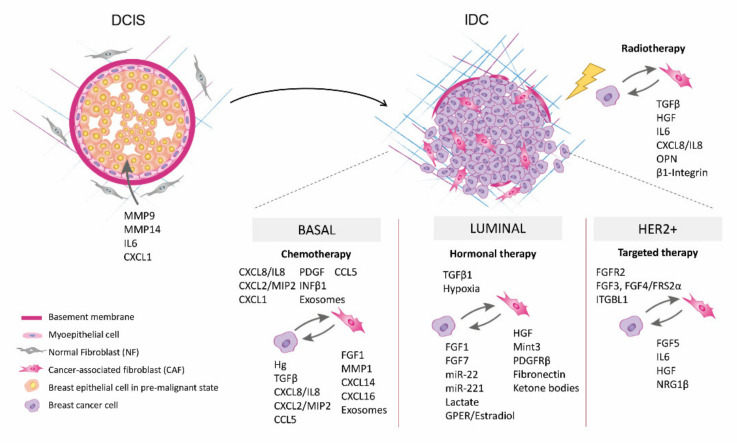
Tumor/stroma communication in ductal carcinoma in situ (DCIS) lesions (**left**) and tumor/stroma crosstalk in invasive breast cancer (**right**) favoring the resistant phenotype of breast cancer cells under radiotherapy, chemotherapy, hormonal and targeted therapy treatments. The main contributors to each situation are listed. For a list of all proteins names abbreviations please see Appendix A.

**Figure 2 cancers-13-03146-f002:**
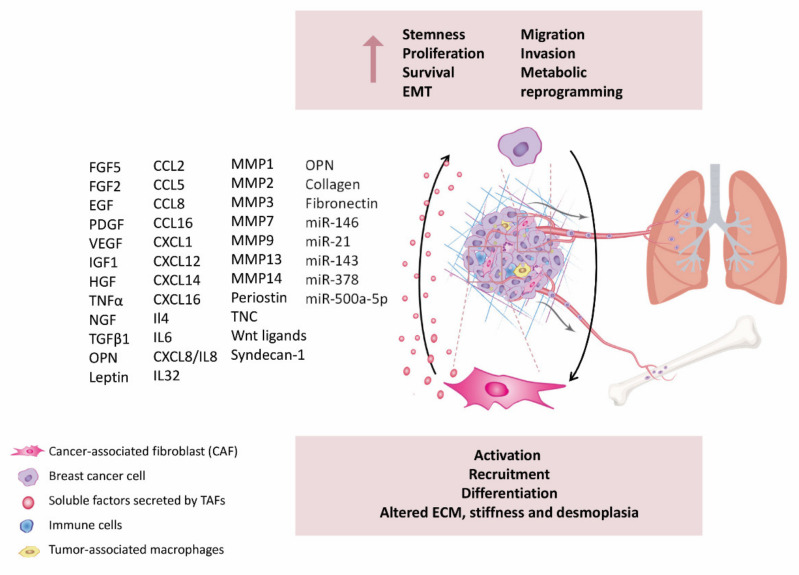
Pro-metastatic CAFs-secreted factors at the primary tumor promote tumor cell migration, invasion, and dissemination. For a list of all proteins names abbreviations please see Appendix A.

**Figure 3 cancers-13-03146-f003:**
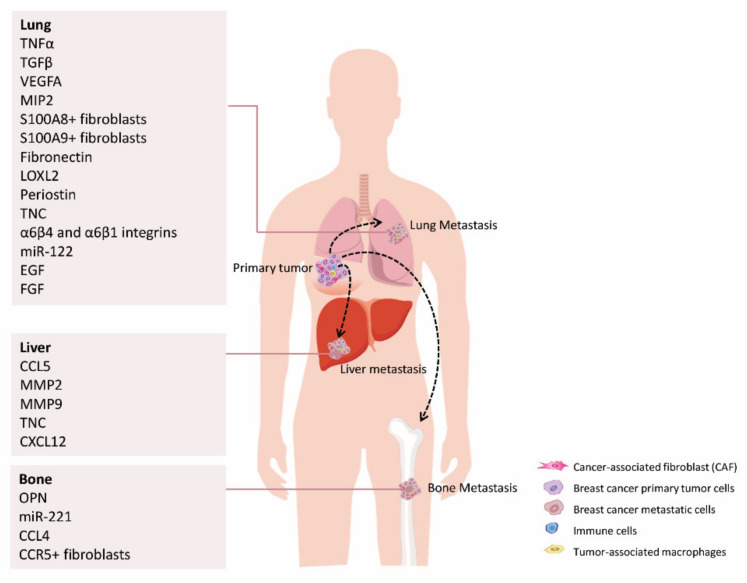
CAFs-secreted factors foster breast cancer cells’ dissemination to major secondary organs, such as lungs, liver and bone. For a list of all proteins names abbreviations please see Appendix A.

## Data Availability

Not applicable.

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
