# Peer review of "Cancer-Associated Fibroblasts in Breast Cancer Treatment Response and Metastasis"

_cancers, 2021, doi:10.3390/cancers13133146_

Round 1
Reviewer 1 Report
This review provides current knowledge on the role of cancer-associated fibroblasts (CAFs) in enhancing breast cancer malignancy. Particularly, the authors reported recent evidences on the involvement of CAFs in sustaining breast cancer progression and metastasis and in modifying the tumor response to the different therapeutic agents (radiotherapy, chemotherapy, hormonal and HER2 target therapy). In addition, an overview of the CAFs-based molecular druggable targets in breast tumors is also provided. Overall the review is well organized and clearly written.
However, some revisions need to be made:
- I would suggest to provide an updated view on the biological and functional properties of CAFs.
- The results of several studies should be summarized using tables since they may represent the most efficient way of presenting results.
- In my opinion the authors need to improve the iconography of figures and to better describe the schematic representation in figure legends
- In the paragraph 4.1 authors should include a comment on the release by CAFs of adipokines, like leptin, that may facilitate the interactions between tumor and stromal cells leading to the malignant aggressiveness.
- The paragraph 5 can be implemented with more clinical trials that have been designed in order to evaluate whether CAFs would be effective targets in breast cancer patients. Moreover, recent preclinical studies describing the impact of nuclear receptors (like FXR) or the enzyme phosphodiesterase PDE5 on the mechanical properties and the paracrine signaling repertoire of CAFs could be reported, since they may represent new potential therapeutic targets.
Reviewer 2 Report
In this manuscript the authors review the current knowledge of cancer-associated fibroblasts (CAFs) in breast cancer. The focus of the manuscript is the role of CAFs in the metastatic cascade and resistance against anti-cancer drugs. The review is well-structured and easy to read. However, some minor points need to be addressed prior to acceptance for publication in Cancers.
Minor comments:
- Simple summary: please explain words like ‘stromal’ and avoid use of words like ‘HER2’.
- Introduction section: rule 60 states ‘There is no targeted therapy against basal or triple negative 60 BrCas (TNBrCa)’. The authors should mention the recent advances in treating BRCA-deficient TNBrCa with PARP inhibitors like olaparib.
- Section 3 is titled: Role of CAFs in BrCa resistance. This should be: Role of CAFs in BrCA therapy resistance.
- There is no consistency in the naming of proteins. Some are written out with abbreviation behind it in brackets and others not. I would suggest keeping the abbreviations in the text and adding a table that explains the abbreviations.
- Section 4.2: title is a bit vague: CAFs from secondary organs microenvironment. I would suggest rephrasing this to: contribution of fibroblasts from metastatic sites to tumor progression.
- Section 4.2: at the end there is a referral to figure 3. However, figure 3 is not in the manuscript.
